# Separation of Bioproducts through the Integration of Cyanobacterial Metabolism and Membrane Filtration: Facilitating Cyanobacteria’s Industrial Application

**DOI:** 10.3390/membranes12100963

**Published:** 2022-09-30

**Authors:** Fei Hao, Xinyi Li, Jiameng Wang, Ruoyue Li, Liyan Zou, Kai Wang, Fuqing Chen, Feixiong Shi, Hui Yang, Wen Wang, Miao Tian

**Affiliations:** 1School of Ecology and Environment, Northwestern Polytechnical University, Xi’an 710072, China; 2Center of Special Environmental Biomechanics & Biomedical Engineering, School of Life Sciences, Northwestern Polytechnical University, Xi’an 710072, China; 3School of Astronautics, Northwestern Polytechnical University, Xi’an 710072, China

**Keywords:** cyanobacterial metabolism, microfiltration, ultrafiltration, bioproducts separation, secreted sucrose

## Abstract

In this work, we propose the development of an efficient, economical, automated, and sustainable method for separating bioproducts from culture medium via the integration of a sucrose-secreting cyanobacteria production process and pressure-driven membrane filtration technology. Firstly, we constructed sucrose-secreting cyanobacteria with a sucrose yield of 600–700 mg/L sucrose after 7 days of salt stress, and the produced sucrose could be fully separated from the cyanobacteria cultures through an efficient and automated membrane filtration process. To determine whether this new method is also economical and sustainable, the relationship between membrane species, operating pressure, and the growth status of four cyanobacterial species was systematically investigated. The results revealed that all four cyanobacterial species could continue to grow after UF filtration. The field emission scanning electron microscopy and confocal laser scanning microscopy results indicate that the cyanobacteria did not cause severe destruction to the membrane surface structure. The good cell viability and intact membrane surface observed after filtration indicated that this innovative cyanobacteria–membrane system is economical and sustainable. This work pioneered the use of membrane separation to achieve the in situ separation of cyanobacterial culture and target products, laying the foundation for the industrialization of cyanobacterial bioproducts.

## 1. Introduction

The continued large-scale utilization of nonrenewable fossil fuels worldwide has led to an alarming level of CO_2_ in the atmosphere, exacerbating global warming and climate change and raising concerns about the shortage of sustainable and eco-friendly resources needed to sustain modern daily life [1,2,3]. To solve these problems, photosynthetic microorganisms, especially cyanobacteria, are receiving increasing attention [4,5,6]. Cyanobacteria can be found in a wide range of ecological habitats on earth and are the only prokaryotes that perform oxygenic photosynthesis [7,8]. Their Simple nutrient requirements (CO_2_, sunlight, and water), rapid cell growth, high photosynthetic efficiency, and ease of genetic manipulation make cyanobacteria an ideal synthetic biology platform for the microbial production of valuable bioproducts [2,9,10]. Therefore, many researchers view cyanobacteria as a promising way of reducing atmospheric carbon levels via fixing CO_2_ and converting it directly into bioproducts [11,12,13,14].

For the industrial application of cyanobacteria, however, many challenges still remain to be overcome. In recent decades, through the use of metabolic engineering and synthetic biology tools, cyanobacteria have been very successful in producing ethanol [15], 2,3-butanediol [16], ethylene [17], limonene [18], 3-hydroxypropionic acid [19], fatty acid [20], lactate [21], astaxanthin [22], and sugars [23], but how to separate these bioproducts from the cyanobacteria is a key economic issue for full-scale industrial production [24,25]. The general method for separating bioproducts from microorganisms is first to break up the cells and then separate the target product via high-speed centrifugation and lysis with an organic reagent [26,27], which is cumbersome, expensive, destructive, and unsustainable. The ideal bioproduct production and separation model should avoid destructive damage to producers and microbial cells, as well as the loss of time and cost for separating products and culturing microorganisms. For this purpose, pressure-driven membrane filtration technologies [28], such as microfiltration (MF), ultrafiltration (UF), nanofiltration (NF), and reverse osmosis (RO), are particularly attractive bioproduct separation methods to apply to industrial processes owing to their cost-effectiveness, high separation efficiency, relatively low environmental impact, and a high degree of automation. These membrane technologies have been effectively used in algae-laden wastewater treatment and water purification [29], and numerous studies have been published on cyanobacteria and related membrane filtration performance. For example, Qu et al. [30] studied the UF membrane fouling caused by cyanobacterial cells and extracellular organic matter (EOM), and they found that EOM could lead to serious irreversible membrane fouling probably caused by protein adhesion, and the fouling was exacerbated when the cyanobacterial cells and EOM were filtered together. Liang et al. [31] demonstrated that humic acid and microbial metabolites are major components of the EOM from two typical cyanobacteria species (*Microcystis aeruginosa* and *Pseudoanabaena* sp.) and also found that EOM could fill the voids of cake layers formed by the algal cells, which indicates that EOM and algal cells play synergistic roles in membrane fouling. Gao et al. [32] investigated the UF fouling behavior of EOMs released from monocultures or different cocultures of cyanobacteria, and they found EOMs from cocultures with a high probability had less flux drop during filtration than monocultures. Zhu et al. [33] posited that the separation performance of UF during the treatment of algae-laden water is highly impacted by the presence of an anionic surfactant.

Although there are many studies on the membrane filtration of cyanobacteria, most of them focus on the membrane fouling behavior caused by the cyanobacteria, aiming to create highly efficient methods of sewage water treatment and municipal drinking water purification. However, few studies have focused on other important research questions regarding the cyanobacteria membrane filtration process, such as the effect of dynamic process parameters and the intrinsic separation characteristics of membrane filtration on cyanobacteria survival, and such studies may have important implications for the application of membrane technology to the separation of bioproducts from cyanobacteria.

In the present study, we conducted MF and UF with two common species of unicellular cyanobacteria, *Synechococcus elongatus* PCC 7942 and *Synechocystis* sp. PCC 6803 (hereafter *Syn*7942 and *Syn*6803, respectively), and two species of classical multicellular cyanobacteria, *Anabaena* sp. PCC 7120 and *Leptolyngbya* sp. strain BL0902 (hereafter *Ana*7120 and *Lep*0902, respectively) in our membrane filtration and bioproduct separation experiments. To obtain a better understanding of the operating mechanism and potential application of the membrane technology for separating bioproducts from cyanobacteria culture, we investigated the interaction between the membrane and cyanobacteria species, the effect of feed pressure on permeate flux, and the separation efficiency and membrane fouling caused by cyanobacteria. Furthermore, we also surveyed the survival of cyanobacterial cells after filtration and demonstrated the feasibility and practicality of membrane separation of bioproducts from cyanobacteria culture using sucrose-secreting cyanobacteria as an example. In summary, this work will provide proof-of-concept support for the application of membrane technology in cyanobacterial synthetic biology and will contribute to the industrialization of cyanobacterial-based carbon recycling and bioproduct production.

## 2. Materials and Methods

### 2.1. Chemicals and Reagents

High-performance liquid chromatography (HPLC) grade acetonitrile was purchased from Merck (Darmstadt, Germany), and deionized water was prepared with Milli-Q water (Millipore Corp., Saint-Quentin, France). Chloramphenicol was purchased from Sangon Biotech Co., Ltd. (Shanghai, China). Sucrose was purchased from Sangon Biotech Co., Ltd. (Shanghai, China). Two commercial membranes, an MF membrane (010) and a UF membrane (050), were purchased from RisingSun Membrane Technology Co., Ltd. (Beijing, China).

### 2.2. Culture Conditions

The cyanobacteria *Syn*7942, *Syn*6803, and *Ana*7120 were purchased from Freshwater Algae Culture Collection at the Institute of Hydrobiology (Chinese Academy of Sciences, Wuhan, China), and *Lep*0902 was obtained from Prof. Xudong Xu of the Institute of Hydrobiology. Unless otherwise specified, all engineered and wild-type cyanobacteria cells were cultured in BG11 liquid medium or on solid agar plates [34] in an illuminating incubator (Crystal, IS-6CL, Dallas, TX, USA) under a constant light intensity of 12,000 lux or illuminating shaking incubator (Crystal, IS-6CL, Dallas, TX, USA) at 30 °C, 100 rpm, under a constant light intensity of 12,000 lux and aeration with 1.5% CO_2_. The antibiotic chloramphenicol (20 µg/mL) was added to the BG-11 growth medium. Cyanobacteria cell growth was monitored by measuring the optical density at 730 nm (OD_730_) with an ultraviolet spectrophotometer (Biochrom, WPA Biowave II, Cambridge, UK). *Escherichia coli* Trans 5a (Tsingke, Beijing, China), used as a host for constructing all recombinant plasmids, was grown on standard Luria–Bertani medium at 37 °C, supplemented with chloramphenicol (100 µg/mL) to maintain the plasmids.

### 2.3. Constructed Plasmids and Strains

All the primers and plasmids used in this study are listed in Appendix A, respectively. The genes and vector fragments used to construct target plasmids were amplified with standard PCR reactions using high-fidelity DNA polymerase (Vazyme, Nanjing, China), and all the template sequences of genes and plasmids are listed in Appendix A. The resulting fragments were assembled using a ClonExpress MultiS One Step Cloning Kit (Vazyme, Nanjing, China) or via Gibson Assembly (New England Biolabs, Ipswich, MA, USA) in accordance with the manufacturer’s instructions. All the primers and target genes in this study were commercially synthesized by Tsingke Biotechnology Co., Ltd. (Beijing, China) and Suzhou Jinweizhi Biotechnology Co., Ltd. (Suzhou, China).

For the transformation of *Syn*7942, 2 mL cells at OD730 = 1 were collected via centrifugation (1000 rpm, 2 min, 4 °C) and washed twice, first with 1 mL of 10 mM NaCl solution and then with fresh BG11 medium. The washed cyanobacteria cells were resuspended in 100 uL of BG11 medium, 5 µg of plasmid DNA was added to the cell mixture, and the mixture was incubated in the dark at 30 °C. The cells were subsequently plated on a BG11 plate containing the appropriate antibiotic for approximately 2 weeks until a single clone appeared. Positive clones were confirmed by PCR verification. The strains used and constructed in this study are listed in Appendix A.

### 2.4. Membrane Filtration Experiment

The filtration experiment was performed in a lab-scale crossflow filtration system in constant pressure mode. A schematic diagram of the membrane filtration system is shown in Figure 1. Briefly, the membrane sample was placed in a test cell with an effective membrane surface area of 34 cm^2^. A peristaltic pump was used to drive the feed solution contained in a feed tank (1 L glass beaker) through the testing unit, with the pressure adjusted by a pressure regulating valve. The concentrated solution refluxed into the feed tank for a continuous filtration cycle, and the permeate was collected into the permeate tank (1 L glass beaker) for measurement of the membrane permeate flux. Two commercial membranes, an MF membrane (010) and a UF membrane (050), were studied, and all the membrane samples were gently rinsed sequentially with an ethanol solution and deionized water before use.

Prior to each membrane filtration experiment, 100 mL of deionized water was filtered to rinse the MF or UF system and stabilize the flux of the membrane. Then, 500 mL of cyanobacteria cultures was poured into the feed tank via a graduated cylinder and filtered under constant pressure. After filtration, the cyanobacteria cultures were sampled from the feed tank for measurement of OD_730_ and the growth curve. The permeate solution in permeate tank was used for measurement of permeate flux via the electronic balance, as shown in Figure 1. After 1.5 min, the permeate flux was named *J*_0_ and then recorded as *J*_i_ at regular intervals. The permeate flux *J* (Lm^−2^h^−1^) can be calculated as:*J* = Δ*m*/(Δ*t*·*ρ*·*A*) (1)
where Δ*m* is the accumulated mass (g) of the permeate solution in the permeate tank for a given time of Δ*t*, *ρ* is the density of the permeate solution (g/mL), and *A* is the effective membrane surface area (m^2^).

As reported in previous studies [30,35], the reversible fouling of the UF membrane by *Syn*7942 was measured. In brief, 200 mL deionized water was filtered with the average permeate flux recorded as *J*p_(0)_. Then the UF membrane was filtered with *Syn*7942 for three continuous filtration cycles. Each filtration cycle included three steps: (1) the UF membrane was filtered using 450 mL *Syn*7942 cultures; (2) the UF membrane was gently washed with deionized water until no obvious cyanobacteria could be removed anymore; (3) the UF membrane was filtered using 200 mL deionized water. The steady flux in each UF filtration cycle using *Syn*7942 cultures was named *Jf*_(n)_. The pure water flux of each UF filtration after wash was named *Jp*_(n)_. The number n represented the cycle number. Then, reversible fouling (***RF***) and irreversible fouling (***IF***) can be calculated as follows:***IF***_n_ = (*Jp*_(0)_ − *Jp*_(n)_)/*Jp*_(0)_
(2)
***RF***_n_ = (*Jp*_(n)_ − *Jf*_(n)_)/*Jp*_(0)_
(3)

### 2.5. Sucrose Assays

For sucrose production, 100 mL cultures of mutant sucrose-secreting cyanobacteria was shocked by the addition of NaCl solutions during the late exponential phase and then continuously grown under the standard culture conditions. At the start of the sucrose assays, 2 mL aliquots of the cultures was centrifuged at 10,000 rpm for 15 min to obtain cell pellets and supernatants. The resulting supernatants were used for sucrose determination by high-performance liquid chromatography (HPLC) (Waters) equipped with a refractive index detector (RID) and a column (250 × 4.6 mm i.d., 5 μm (Spherisorb@ NH_2_ (amino), Waters, Milford, MA, USA) maintained at 35 °C [36]. The mobile phase consisted of acetonitrile: water (75: 25, *v*/*v*) was degassed by ultrasonic bath before use. Each run was completed within 30 min. The flow rate was 0.6 mL min^−1^, and a 20 uL aliquot of sample solution was injected into the HPLC–RID system. All samples and standards were filtered through a 0.45 µm Millipore membrane before use.

### 2.6. Fouled Membrane Characterization

Cyanobacteria-induced membrane fouling was characterized using field emission scanning electron microscopy (HITACHI Regulus 8100, Tokyo, Japan). After membrane filtration was completed, the membrane was gently rinsed with deionized water to remove the surface-adhering dirt. The fouled membrane was then cut into pieces (2 cm × 2 cm) and immersed in a fixative solution. Finally, the fixed samples were sent to a Chinese company Servicebio for field emission scanning electron microscopy (FESEM) experiments. The membrane surface morphology was imaged by a confocal laser scanning microscopy (CLSM, Nikon, model AX R, Tokyo, Japan). A 561-nm laser was used for the excitation of all samples. The emission filter had a wavelength range of 570–620 nm. After being rinsed with deionized water, the fouled membrane was cut into small pieces and then placed in glass-bottomed dishes for observation by confocal laser scanning microscopy (CLSM) under a 40× objective.

## 3. Results and Discussion

### 3.1. Membrane Fouling Behavior of Different Kinds of Cyanobacteria

The cyanobacteria species and the operating pressure used in the process of bioproduct separation from cyanobacteria can affect the separation efficiency achieved by this process. We chose four cyanobacteria species to use in our evaluation of membrane filtration efficiency and retention. These four cyanobacteria species were first used in research on membrane filtration technology. The two unicellular cyanobacteria species (*Syn*7942 and *Syn*6803) have both been reported to successfully produce many valuable bioproducts [6,7,24]. Additionally, *Syn*7942, *Syn*6803, and *Ana*7120 are also common freshwater cyanobacteria in rivers and lakes [37,38,39].

The normalized fluxes of the four species of cyanobacteria at different test pressures over time are presented in Figure 2. These results show that the type of cyanobacteria solution had a minor effect on the decline of the permeate flux, and all fluxes decreased rapidly in the initial stage, after which they gradually became stable. The observations are also consistent with the findings of previous studies [31]. As shown in Figure 2a–d, when the feed pressure was increased from 0.34 bar to 1.0 bar, the permeate fluxes of the filtration for each of the four species of cyanobacteria decreased more quickly. Moreover, the flux decline curves of *Syn*6803 and *Syn*7942 were relatively close throughout the filtration process but slightly dispersed from those of *Ana*7120 and *Lep*0902, which implies that the permeate flux declines of *Ana*7120 and *Lep*0902 filtration were more affected by the feed pressure compared with those of *Syn*6803 and *Syn*7942. Notably, *Syn*6803 (~1.5 μm wide, ~1.5 μm long) [40] and *Syn*7942 (~1.5 μm wide, ~3.5 μm long) [41] are both species of unicellular cyanobacteria, and *Ana*7120 (~1 μm wide, ~1.3 μm long) [42] and *Lep*0902 (~1.5 μm wide, ~3.6 μm long) [43] are both species of multicellular cyanobacteria. These results suggest that the cell morphology of the cyanobacteria and feed pressure may synergistically affect the filtration performance of cyanobacteria.

### 3.2. Cyanobacteria Survival after Pressure-Driven Membrane Filtration

To study the influence of the membrane filtration process on cyanobacteria survival, we surveyed the cell growth of cyanobacteria after filtration. Specifically, 500 mL of cyanobacteria cultures with OD_730_ = 1.5–1.8 was filtered by an MF membrane at a constant pressure of 0.34 bar, 0.69 bar, or 1.00 bar. When the MF filtration was finished, 100 mL of the circulating filtered cyanobacteria cultures in the feed tank was taken out into a 250 mL flask and directly cultured in the same condition as before. Then 10 mL of the circulating filtered cyanobacteria cultures in a feed tank was inoculated into a 250 mL flask containing 100 mL of fresh BG11 medium after MF filtration and also cultured in the same condition as before. As shown in Figure 3a–f, whether they were directly cultured or inoculated into fresh BG11 medium, *Syn*6803, *Ana*7120, and *Lep*0902 all maintained continuous growth rates after the MF process under various feeding pressures, which means that the MF process did not seriously affect their growth. In contrast, the *Syn*7942 cultures became bleached within 4 days after MF and exhibited no signs of growth (Figure 3g,h), suggesting that the *Syn*7942 cells might have suffered irreversible damage under pressure.

To determine the factors that affect the survival and growth of *Syn*7942 during the MF process, the feed pressure, filtration time, and membrane type were studied. The constant feed pressures of 0.34 bar and 0.1 bar were tested, with 10 mL of *Syn*7942 cultures collected and inoculated into 100 mL of fresh BG11 medium at the time points of 5 min, 15 min, 25 min, 35 min, and 45 min from the start of the filtration. Even at a feed pressure as low as 0.1 bar (the lowest operating pressure for collecting permeate) applied for only 5 min of MF, the re-cultured *Syn*7942 cells entered a decline phase after only 1 or 2 days of slight growth (Figure 4a,b). However, when using a UF membrane for filtration with a pressure of 1 bar, the four species of cyanobacteria cultures all showed continuous growth (Figure 4c,d). This suggests that the feed pressure and membrane pore structure synergistically affected the survival condition of *Syn*7942 cells after filtration, and an MF membrane with its larger void area may cause more damage to *Syn*7942 cells compared with a UF membrane. Thus, these results indicate that the growth of cyanobacteria may be seriously affected during a pressure-driven membrane filtration process, and the resistance to environmental stresses of cyanobacteria varied significantly. *Syn*6803, *Ana*7120, and *Lep*0902 exhibited strong compressive ability, whereas *Syn*7942 was very sensitive to compressive stress. Even a short period of MF filtration may cause fatal irreversible damage to *Syn*7942 cells. On the other hand, using a UF membrane can eliminate the fatal effect of filtration on cell activity.

Previous literature [44,45] reported that multicellular organisms have superior resistance to environmental stresses compared with unicellular organisms. Additionally, when cultured in a liquid medium, round-shaped *Syn*6803 cells will cluster into a compact regiment, whereas rod-shaped *Syn*7942 cells will take on filamentous morphology [46,47,48]. Maybe these reasons can explain why the MF filtration did not seriously affect the cell survival of *Syn*6803, *Ana*7120, and *Lep*0902. The study of the impact of the membrane filtration process on cyanobacteria cells survival is also first reported here, as most previous studies mainly showed concerns about the fouling behavior of cyanobacteria on the membrane [32,33,35,49,50].

To figure out whether the cyanobacteria can pass through the MF or UF during filtration, 5 mL permeate solution in the permeate tank was taken out into a 250 mL flask containing 100 mL of fresh BG11 medium after MF or UF filtration of the four cyanobacteria species and cultured in the same condition of culturing cyanobacteria. The result of monitoring OD_730_ (Appendix A) showed there were no cyanobacteria cells in the permeate solution.

### 3.3. Characterization of Fouled Membranes

After finding that the post-MF cell survival condition of *Syn*7942 was completely different from that of UF, we speculated that the interaction between the MF or UF membrane surface properties and *Syn*7942 cells might be responsible for this difference. Therefore, we used CLSM and FESEM to examine the properties of MF membrane surface pores and the attachment of *Syn*7942 cells on the membrane surface after filtration. As shown in the CLSM images (Figure 5 and Appendix A), when the imaging layers were 5–35 μm depth from the surface of the membrane, the visible membrane pores of UF were smaller than that of MF, and all of them were intact, indicating that the structure of the MF and UF membrane was not greatly affected by filtration with *Syn*7942 cells. The FESEM results, shown in Figure 6a,b, reveal that, while there were a few intact *Syn*7942 cells present on the MF membrane surface, a considerable amount of residue that could be cell debris was also present. However, a large cake layer formed by many *Syn*7942 cells, cell debris, and EOMs could be seen on the UF membrane surface (Figure 6c,d). Both commercial membranes, MF and UF, are hydrophilic membranes, and the MF membrane has a smaller contact angle than the UF membrane (Appendix A). Moreover, the pore size test results using a porometer (BSD-PB, BeiShiDe Instrument) showed that the MF010 membrane has a mean pore size of 95.4 nm, and the UF050 membrane has a mean pore size of 16.6 nm (Appendix A). Therefore, compared with the UF membrane, the cell debris and EOM are easier to enter the MF membrane pores owing to the bigger size, which may make it difficult to form a cake layer on the MF membrane surface. It has been reported that the membrane filtration process will produce hydrodynamic shear forces, which can break the cyanobacteria cells [51]. Many previous studies [30,31,33,35] have found that a cake layer will quickly form on the UF membrane surface when UF filtration of cyanobacteria starts, and such cake layer can prevent the cyanobacteria cells from entering the UF membrane pores, which may also protect the cyanobacteria cells from the damage caused by hydrodynamic shear force during the filtration process and the sharp edges of large pores. Thus, in comparison with the UF filtration, there are more *Syn*7942 cells that appear to have been damaged when using MF filtration, which produced a large amount of cell debris and toxic intracellular metabolites in a short time, preventing the *Syn*7942 cells from growing. The cake layer can also prevent the membrane surface from being directly bombarded by cyanobacteria cells, and that is why we can see from Figure 6 that the UF membrane has a more intact surface structure than the MF membrane after cyanobacteria filtration.

### 3.4. Construction of cscB^+^ Syn7942 and the Separation of Produced Sucrose

Without the addition of a sucrose transporter gene, the obligate photoautotrophic organism *Syn*7942 cannot assimilate or export sucrose [52,53]. However, *Syn*7942 is able to synthesize and accumulate cytoplasmic sucrose when the external osmotic pressure exists [23,54]. Previous studies have demonstrated that some cyanobacteria can secrete intracellular sucrose into the culture medium during salt stress via the heterologous expression of the *E. coli* sucrose permease-encoding gene *cscB* [23,25]. Therefore, the *cscB* gene (*ECW_m2594*) driven by the constitutive promoter *PpsaAB* from *Syn*6803 was integrated into the NS1 site of the genome of *Syn*7942 (Figure 7a), which can tolerate such an insertion with no phenotypic effects, in order to create the sucrose-secreting cyanobacteria Tcya-1. The result of colony PCR of Tcya-1 (Appendix A) demonstrates that the integration of the *cscB* gene into the *Syn*7942 chromosome was successful, resulting in the *cscB*^+^ *Syn*7942. The growth of Tcya-1 and wild-type *Syn*7942 were compared (Figure 7b), and different concentrations of NaCl (0 mM, 100 mM, 150 mM, and 200 mM) were added into cyanobacteria cultures with OD_730_ = 1 to test their sucrose export rates (Appendix A). Additionally, Tcya-1 was able to secrete sucrose into the culture supernatant under salt stress, unlike the wild-type *Syn*7942, which lacked *cscB* expression and was unable to secrete sucrose (Figure 7c). The most efficient secretion of sucrose was observed when 150 mM NaCl was added (Figure 7c and Appendix A); this led to the production of 600–700 mg/L sucrose after 7 days of salt stress (Figure 7c and Appendix A), which is comparable to the levels reported in a previous study [25].

To test the application of a membrane filtration system on the separation of sucrose from cyanobacteria cultures, the cells in 500 mL of Tcya-1 culture (OD_730_ = 1) were shocked by the addition of a 150-mM NaCl solution. As the level of produced sucrose reached 600 mg/L, the 500 mL of Tcya-1 culture was moved into a feed tank to undergo filtration with an MF or UF membrane under the constant pressure of 1 bar. After the filtration was finished, the level of sucrose in the permeate solution in the permeate tank was determined. The results shown in Figure 7d demonstrate that the sucrose concentration of the permeate solution was 600 mg/L (i.e., the same as the concentration prior to filtration), which means that the extracellular sucrose produced by cyanobacteria can be fully separated by a membrane filtration system without any loss. Since the evaluation of repeatable use of a membrane is a very important aspect from an application point of view, the reversibility of UF membrane fouling of filtering *Syn*7942 was analyzed. As shown in Figure 8a, when the third filtration cycle started, the initial flux dropped by nearly 50%, while the steady flux did not change significantly. The results in Figure 8b showed *Syn*7942 could cause both reversible and irreversible fouling during UF filtration, and the irreversible fouling showed a slight increase as more filtration cycles were carried out.

Because our study results show that the cyanobacteria could continue to grow after filtration and that the membrane surface structure was not seriously damaged during filtration, the cyanobacteria–membrane system described in this work could be used to culture cyanobacteria for a long time while simultaneously separating their bioproducts, in a manner similarly sustainable as milk production by cows.

Here, through the integration of a sucrose-secreting cyanobacteria production process and pressure-driven membrane filtration technology, we created an innovative bioproduct separation method that can efficiently, economically, automatically, and sustainably yield sucrose, which is a commonly used carbon source for chemicals and food production [25] and can enhance the practical applications of cyanobacteria biosynthesis for CO_2_ capture and carbon neutral production. In addition, this work will also provide a fundamental basis for further optimization of the cyanobacteria–membrane system for bioproduct production and separation. For example, to potentially discover a more suitable method of cyanobacteria bioproduct separation, future work could: (1) try transforming the feed tank into a controllable cyanobacteria culture device for cyanobacteria growth while separating bioproduct model testing with pressure-driven membrane filtration technology, and (2) conduct an exploration of nanofiltration (NF) and reverse osmosis (RO) use. By conducting such optimization studies, we hope our research can contribute to the creation of ideal cyanobacteria bioproduct production and separation model that is able to meet the chemical and food supply needs of humans and also avoid CO_2_-induced environmental problems.

## 4. Conclusions

This work systematically investigated the relationship between membrane type, operating pressure, and the growth status of four different species of cyanobacteria (*Syn*7942, *Syn*6803, *Ana*7120, and *Lep*0902) to create an innovative bioproduct separation method that integrates a sucrose-secreting cyanobacteria production process and membrane filtration technology. The results showed that the morphologies of cyanobacteria led to differences in their filtration performance when the feed pressure was varied. The permeate flux decline was more affected by feed pressure in multicellular cyanobacteria than in unicellular cyanobacteria. We also found that the cyanobacteria species, feed pressure, and membrane pore structure synergistically affected the survival condition of cyanobacteria cells after filtration. *Syn*6803, *Ana*7120, and *Lep*0902 could continue to grow after filtration using an MF membrane, whereas *Syn*7942 could not, owing to irreversible damage caused during filtration with an MF membrane though under low pressure. However, *Syn*7942 could continue to grow after filtration when a UF membrane was used instead, even at a higher pressure. Moreover, the cake layer formed on the UF membrane surface may protect the cyanobacteria cells from the damage caused by the hydrodynamic shear force and prevent the membrane surface from being directly bombarded by cyanobacteria cells during the filtration process, which allows the *Syn*7942 to continue to grow after filtration. At last, the genetically modified *cscB*^+^
*Syn*7942 (Tcya-1) could produce 600–700 mg/L sucrose after 7 days of salt stress. The produced sucrose could be fully harvested through filtration with an MF or UF membrane and achieved real-time 100% recovery of *cscB*^+^
*Syn*7942 (Tcya-1).

In summary, using the schematic diagram shown in Figure 1 as a reference with UF filtration of cyanobacteria, there will appear a promising cyanobacteria bioproducts separation model with the potential to be automated, sustainable, economical, and efficient.

## Figures and Tables

**Figure 1 membranes-12-00963-f001:**
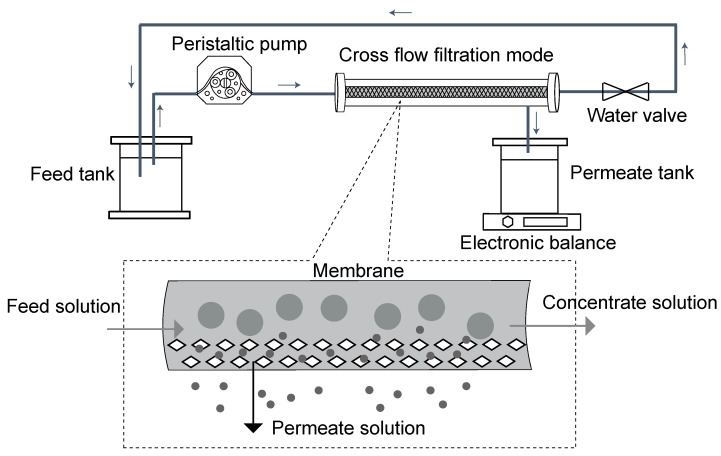
Schematic diagram of the membrane filtration system.

**Figure 2 membranes-12-00963-f002:**
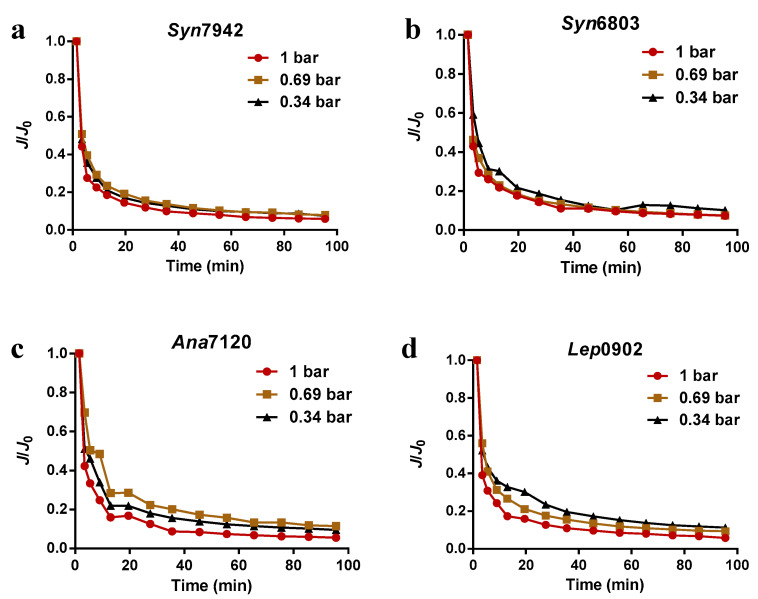
The normalized fluxes of the four species of cyanobacteria at different test pressures over time; (**a**) *Syn*7942; (**b**) *Syn*6803; (**c**) *Ana*7102; (**d**) *Lep*0902. The membrane investigated here was an MF (010), and the OD730 of cyanobacteria used for filtration was 1.5–1.8.

**Figure 3 membranes-12-00963-f003:**
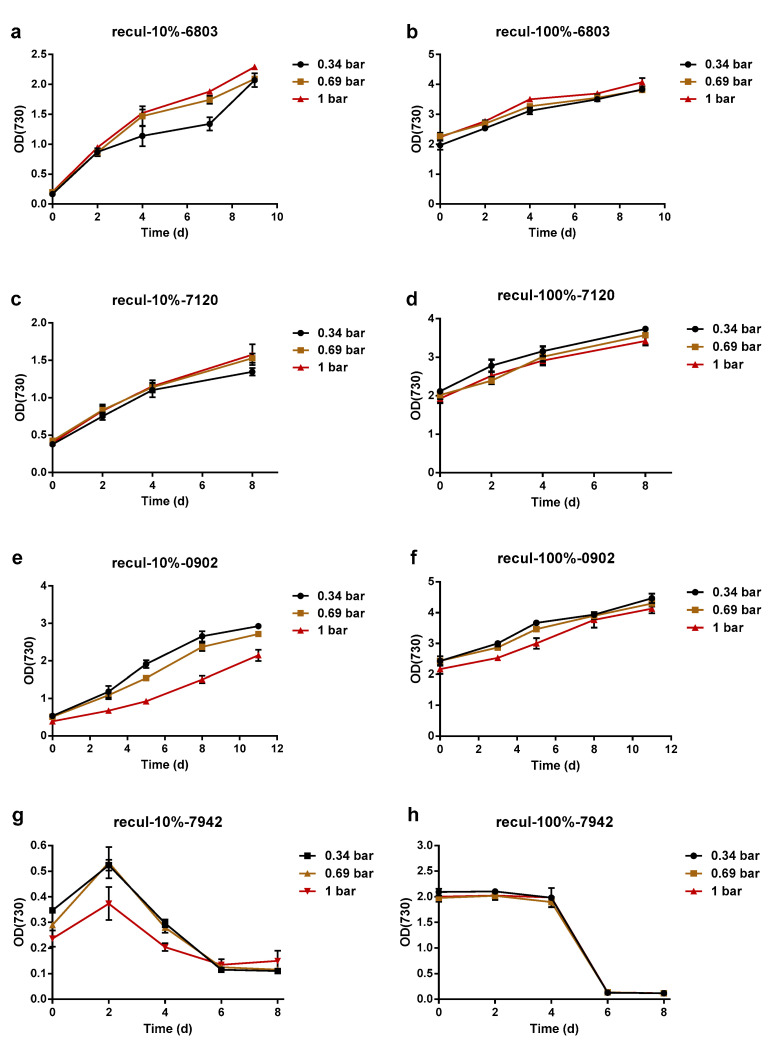
Cyanobacteria growth curves after filtration using an MF membrane; growth curves for *Syn*6803 (**a**,**b**), *Ana*7120 (**c**,**d**), *Lep*0902 (**e**,**f**), and *Syn*7942 (**g**,**h**). For the experiments whose data are shown in (**a**,**c**,**e**,**g**), 10 mL of circulating filtered cyanobacteria cultures was inoculated into 100 mL of fresh BG11 medium for re-culture after 95.5 min of filtration. For the experiments whose data are shown in (**b**,**d**,**f**,**h**), 100 mL of circulating filtered cyanobacteria cultures was directly used for re-culture after 95.5 min of filtration. Error bars represent the standard deviations of triplicates.

**Figure 4 membranes-12-00963-f004:**
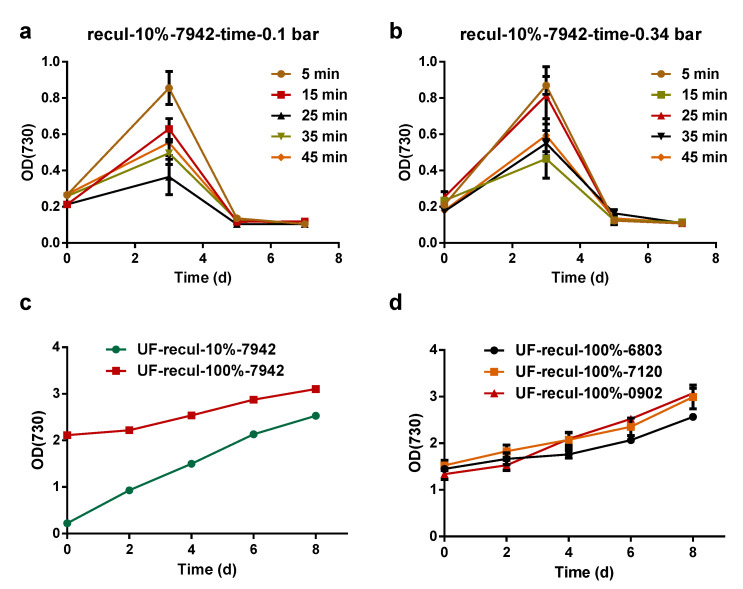
The cyanobacteria growth curve after membrane filtration. (**a**,**b**) For these experiments, 10 mL of circulating filtered *Syn*7942 culture was inoculated into 100 mL of fresh BG11 medium for re-culture after regular intervals of filtration using an MF membrane under a feed pressure of 0.1 bar (**a**) or 0.34 bar (**b**). (**c**) For this experiment, 10 mL of circulating filtered *Syn*7942 culture was inoculated into 100 mL of fresh BG11 medium for re-culture, or 100 mL of circulating filtered *Syn*7942 culture was directly used for re-culture after 95.5 min of filtration using a UF membrane under a feed pressure of 1 bar. (**d**) For this experiment, 100 mL of circulating filtered *Syn*6803, *Ana*7120, and *Lep*0902 culture was directly used for re-culture after 95.5 min of filtration using a UF membrane under a feed pressure of 1 bar. The OD730 of cyanobacteria used for filtration was 1.5–1.8. Error bars represent the standard deviations of triplicates.

**Figure 5 membranes-12-00963-f005:**
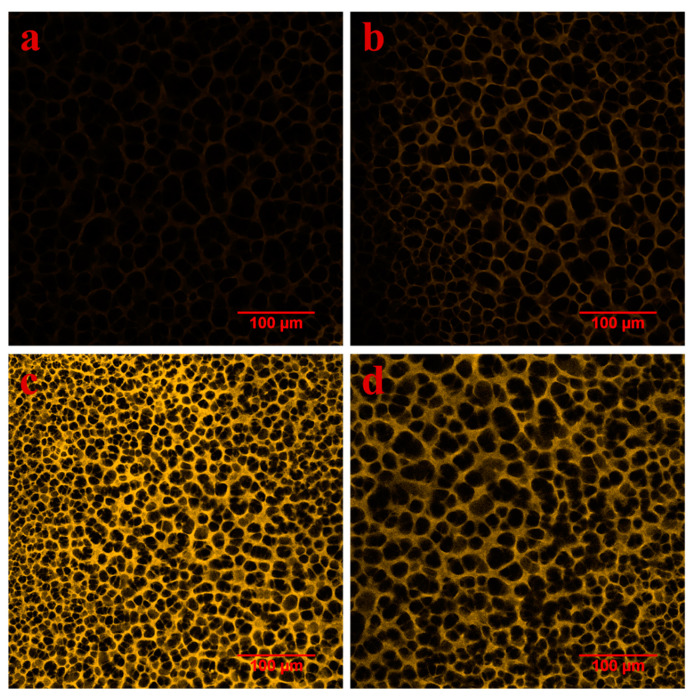
CLSM images of MF or UF membranes. (**a**,**b**) Image of the surface of MF membranes after they were used to filter *Syn*7942 cells under a feed pressure of 0.34 bar (**a**) or 1.00 bar (**b**). (**c**,**d**) Images of the surface of UF membranes after they were used to filter *Syn*7942 cells under a feed pressure of 0.34 bar (**c**) or 1.00 bar (**d**). The imaging layer is 20 μm depth from the surface of the MF or UF membrane.

**Figure 6 membranes-12-00963-f006:**
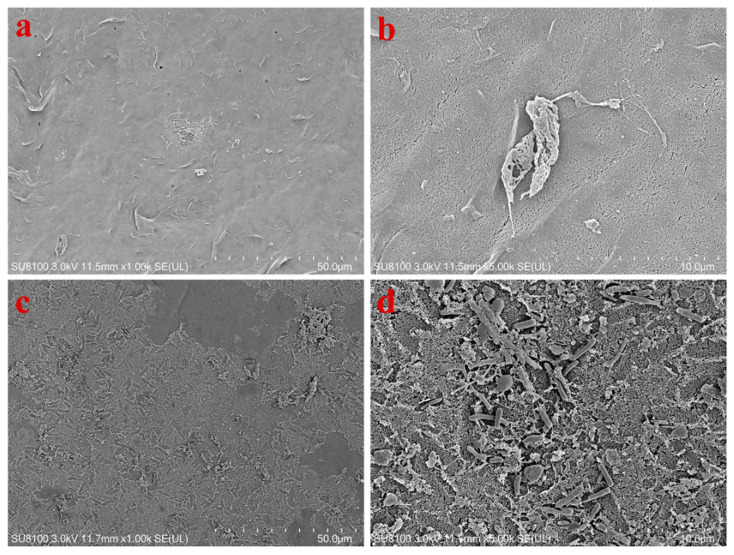
FESEM images of the surface of MF membranes. (**a**,**b**) Images of MF membrane fouled by *Syn*7942 under a feed pressure of 1.00 bar; (**c**,**d**) images of UF membrane fouled by *Syn*7942 under a feed pressure of 1.00 bar.

**Figure 7 membranes-12-00963-f007:**
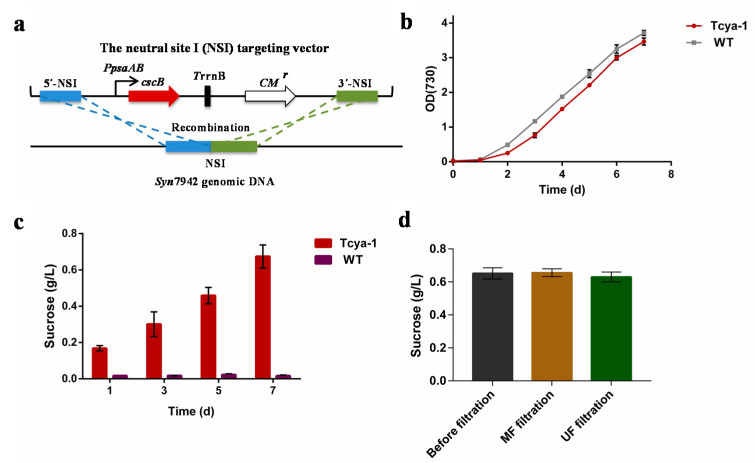
The separation efficiency of biosynthesis filtration system. (**a**) Schematic representation of sucrose transporter gene *cscB* with constitutive promoter *PpsaAB* integration into the *Syn*7942 genome. The *T*rrnB and *CM* represent trrnb transcription terminator and chloramphenicol resistance gene, respectively; (**b**) growth curves of Tcya-1 and wild-type *Syn*7942 (WT); (**c**) Sucrose yields of Tcya-1 and wild-type *Syn*7942 (WT); (**d**) the level of sucrose from cyanobacteria cultures before and after filtration with an MF or UF membrane under a feed pressure of 1 bar. Error bars represent the standard deviations of triplicates.

**Figure 8 membranes-12-00963-f008:**
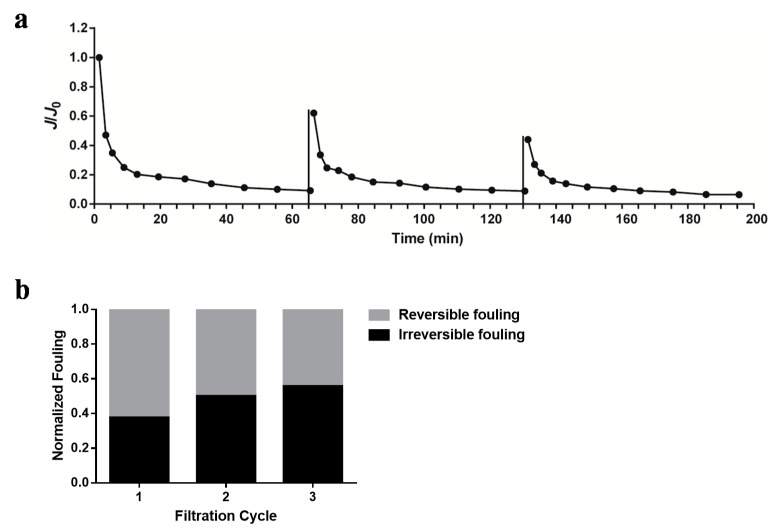
Flux decline (**a**) and reversibility (**b**) of fouling during UF filtration of Syn7942.

## Data Availability

Data will be made available on request.

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
