# Peer review of "Separation of Bioproducts through the Integration of Cyanobacterial Metabolism and Membrane Filtration: Facilitating Cyanobacteria’s Industrial Application"

_membranes, 2022, doi:10.3390/membranes12100963_

Round 1

Reviewer 1 Report

This study investigated the effect of membrane filtration on the cyanobacteria separation, where a few important operation factors and the separation efficiency were evaluated. The results are interesting, however, the writing of this study should be significantly improved before the manuscript could be accepted by the journal.

The detailed comments can be see below:

The abstract is too long and redundant, which should be significantly revised in a concise manner.

The aims of this study written in the introduction section was confusing, which was not well consistent with the title of this manuscript.

How the authors inject the cyanobacteria into the feed and how the samples were taken should be given in detail.

What is the pore size of the MF and UF used in this study?

Fig.5 was not well interpreted, why all the figures except for Fig.5 were completely black, did the author stained the sample?

Line 225-226, Where did the authors take the sample during the filtration, feed tank, permeate tank or filtration module? Can the cyanobacteria pass through the MF or UF and how many of them could pass through?

Line 298-299, We hypothesized that the cake layer on the membrane surface can effectively protect the cyanobacteria cells from damage of hydrodynamic shear force during filtration process. Such hypothesis came from no where, references or other evidence should be provided to prove this hypothesis.

Author Response

We appreciate the comments and suggestions from reviewers and editors. The manuscript has been revised accordingly, and point-by-point responses to the comments are given in the attached document.

Reviewer 2 Report

This study examines the performance of four types of algae under pressurized MF filtration. The manuscript is well written, and the authors also provide some interesting results. The manuscript can be further improved should the following comments can be addressed.

1. The authors find that some algae cannot survive under high pressure, while some can. For the dead algae under high pressure filtration, will they release anything during the filtration? Can the authors provide the organics concentration in the algae solution before and after low/high pressure filtration?

2. What are the four types of algae cells concentration in the MF and UF filtration in this study?

3. The permeate flux decreased significantly in the MF filtration. It seems that the pressure difference didn’t affect the flux much. So in the MF/UF filtration, only low pressure shall be enough and no need to adopt high pressure?

4. The authors measure the colony PCR in this study, however, there was very limited discussion on the measurement. The authors shall provide more discussion on this part of results.

5. What is the pore size and particle size of the algae used in this study?

6. The NaCl can secrete sucrose yield of the algae, what is the mechanism? Can authors provide the secreted sucrose yield information without NaCl dosage?

Author Response

(The authors gave the same response as above.)

Reviewer 3 Report

Review of paper ‘Separation of bioproducts through the integration of cyanobacterial metabolism and membrane filtration’ prepared by Fei Hao, Xinyi Li, Jiameng Wang, Ruoyue Li, Liyan Zou, Fuqing Chen, Feixiong Shi, Hui Yang, Wen Wang and Miao Tian.

Manuscript membranes-1887230 is focused on the presentation of a method based on the sucrose-secreting cyanobacteria production process and the micro- and ultrafiltration separation process. I have some suggestions that authors may consider prior to publishing this work.

1. The processing of CO2 by cyanobacteria is well known, as the authors have demonstrated in the Introduction and in the cited literature. However, what is missing is information indicating the novelty of their own research. This should be clearly emphasised in the paper.

2. Membrane separation depends to a large extent on the selection of a suitable membrane. Therefore, the authors should provide a more detailed characterisation of the membranes used.

3. The authors wrote ‘We speculate that the different cell morphology of the cyanobacteria led to their different filtration performance under different feed pressures’. This needs to be clarified. Would it then be possible to select membranes according to microorganisms without filtration tests but only on the basis of their morphology? Are these assumptions supported by investigations? How can this effect be explained?

4. When evaluating filtration, an important aspect is the evaluation of the process performance, i.e., the flux values obtained during the process. This is a very important aspect from an application point of view. I recommend adding such information to the work.

5. Membrane fouling is an important aspect of the filtration process. The authors should show studies indicating whether membranes can be reused in the filtration process. Are significant decreases in permeation efficiency observed during repeated use of the membrane? How was their cleaning carried out?

6. In the last paragraph of the conclusions, the authors indicated that 'In a word, this work demonstrated that the integration of a sucrose-secreting cyanobacteria production process and pressure-driven membrane filtration technology has promising potential for the efficient, economical, automated, and sustainable separation of cyanobacteria bioproducts'. In my opinion, a proposed installation scheme should appear in the publication, together with a statement of the optimum operating conditions.

7. The figures are blurred. The quality should be improved, in particular Figs. 1-3.

8. The abbreviations used in Figure 7a should be explained.

Author Response

(The authors gave the same response as above.)

Round 2

Reviewer 1 Report

The authors have addressed all of my concerns. The current form of manuscript could be accepted

Reviewer 3 Report

The authors revised the manuscript significantly improving its quality. I have no additional comments.